# Reduced social distancing early in the COVID-19 pandemic is associated with antisocial behaviors in an online United States sample

Katherine O'Connell[1]*, Kathryn Berluti[2], Shawn A. Rhoads[2], Abigail A. Marsh[2]

**1** Interdisciplinary Program in Neuroscience, Georgetown University, Washington, DC, United States of America, **2** Department of Psychology, Georgetown University, Washington, DC, United States of America

* kmo52@georgetown.edu

## Abstract

Antisocial behaviors cause harm, directly or indirectly, to others' welfare. The novel coronavirus pandemic has increased the urgency of understanding a specific form of antisociality: behaviors that increase risk of disease transmission. Because disease transmission-linked behaviors tend to be interpreted and responded to differently than other antisocial behaviors, it is unclear whether general indices of antisociality predict contamination-relevant behaviors. In a pre-registered study using an online U.S. sample, we found that individuals reporting high levels of antisociality engage in fewer social distancing measures: they report leaving their homes more frequently (p = .024) and standing closer to others while outside (p < .001). These relationships were observed after controlling for sociodemographic variables, illness risk, and use of protective equipment. Independently, higher education and leaving home for work were also associated with reduced distancing behavior. Antisociality was not significantly associated with level of worry about the coronavirus. These findings suggest that more antisocial individuals may pose health risks to themselves and their community during the COVID-19 pandemic.

## Introduction

The most urgent public health issue of the 21st century thus far is the global COVID-19 pandemic caused by a novel coronavirus strain, estimated to have caused over 1.5 million deaths in 2020 [1]. Because no cure or effective vaccine for COVID-19 is widely available yet, the primary means of reducing illnesses, deaths, and other costs from the pandemic are behavioral. These include social distancing measures such as limiting non-essential trips outside of the home and maintaining adequate social distance (6 feet or greater) from others in public settings [2]. Despite high global awareness of the pandemic and its impact particularly on vulnerable populations like the elderly and those with chronic health conditions [3,4], avoidance of harmful behaviors remains inconsistent [5,6], contributing to the ongoing spread of the virus. Thus, better understanding the factors that promote versus inhibit disease transmission risk behaviors is essential. In light of evidence that antisociality—the tendency to engage in various behaviors that directly or indirectly harm the welfare of others—represents a stable phenotype

**Data Availability Statement:** All survey materials, pre-registration materials, non-identifiable data and statistical code are publicly available on OSF (https://osf.io/3429d/, 10.17605/OSF.IO/3429D).

**Funding:** This research was carried out using internal Georgetown University funding to AAM. This research was also supported by the National Center for Complementary and Integrative Health of the National Institutes of Health under award number F31AT010423 to KO. The funders had no role in study design, data collection and analysis, decision to publish, or preparation of the manuscript.

**Competing interests:** The authors have declared that no competing interests exist.

[7] that varies across individuals and predicts behaviors across domains including physical aggression, social aggression, and rule-breaking [8–12], we predicted that variation in social distancing behavior would be associated with scores on a validated measure of general antisociality even after accounting for key demographic and risk-related variables.

Antisocial behavior is defined as any action that harms others, violates social norms, or infringes on the rights of others [10,13]. The tendency to engage in antisocial behaviors such as violence, rule-breaking, and bullying varies significantly across the population, with a small proportion of individuals responsible for the majority of serious antisocial acts. For example, large cohort studies estimate that the most antisocial 1–10% of the population is responsible for more than two thirds of all criminal convictions [14–16], and 5% of the population is responsible for almost half of all lying [17]. Engagement in antisocial behaviors has been associated with behaviors specifically relevant to public health risks as well, including physical violence [18–21], unsafe driving [22,23], and risky sex practices [24–27]. A relatively small fraction of individuals engaging in disease transmission behaviors could have significant implications, as epidemiological research suggests a small proportion of individual disease hosts can account for massive numbers of cases [28,29].

At the time of this investigation—early April of 2020—COVID-19 had become recognized as a significant cause of serious illness and death in the United States and other countries. Confirmed U.S. cases increased 112% from 186,101 on April 1st to 395,011 on April 8th [30] (the day of data collection), community spread was known to be present in at least 31 U.S. states [30], and general uncertainty regarding risks and illness transmission coincided with widespread closures of schools, restaurants, gyms, and offices. During this time, physical distancing recommendations and stay-at-home/ safer-at-home policies became pervasive and affected 94% of the US population [31,32]. Behaviors that violated these guidelines and increased risk of disease transmission quickly came to qualify as forms of antisociality. Such behaviors included leaving the home and venturing into public spaces for non-essential reasons, and maintaining insufficient distance (less than 6 feet) from others in public settings [33]. It is important to recall that in this early phase of the U.S. pandemic, access to personal protective equipment (PPE), including face masks, was severely limited and messaging regarding the efficacy of face masks for the public was inconsistent [34,35]. U.S. public health organizations recommended *against* the general public wearing masks throughout March (in part due to PPE shortages affecting healthcare workers) and only revised this stance to recommend cloth face coverings for the public on April 3rd, 2020 [35,36]. We therefore focused on non-compliance with social distancing guidelines (but not PPE use) including leaving the home and standing close to others, both of which contributed to spread of the virus and clearly violated norms at the time.

Engagement in these potentially disease-transmitting behaviors remained common in the U.S. even at this time of exponential spread, stay-at-home orders and generally high uncertainty about the virus. Late March/early April polls reported that 16% of Americans were not avoiding social gatherings [37] and photographs of "coronaparties," protests, and crowded beaches led news headlines across the country. Potential reasons for continuing to engage in disease-transmitting behaviors during a pandemic include low awareness of disease severity (which may result from contradictory or unclear public messaging), perceptions of low personal risk, and socioeconomic factors [38–40]. But the antisocial behavior literature makes clear that individual variation in personality and values likely also plays a role [41–45]; in particular, individual variation in overall antisocial tendencies may represent an important contributor to social distancing behaviors during the novel coronavirus pandemic. However, no empirical link between general levels of antisociality and behaviors that risk transmission of the novel coronavirus yet exists. And some evidence suggests disease risk behaviors may be

moderated by dissociable psychological mechanisms from those that moderate other forms of risk behavior, with disease transmission risk behaviors regulated by neurocognitive systems that generate disgust in response to pathogen cues [46–48], and aggressive and other antisocial behaviors regulated by neural systems that generate fearful or angry responses to acute harm [49–51].

We thus sought to empirically test whether, in light of widespread awareness that behaviors that risk coronavirus transmission may expose others to acute illness or death, continuing to engage in such behaviors would be associated with overall antisocial tendencies. We hypothesized that antisociality would correspond to reduced social distancing behaviors. To test this prediction, we recruited an online sample of adults and assessed demographic variables, self-reported social distancing behaviors, and information related to illness risk of both respondents and members of their households. We measured antisociality using the Subtypes of Antisocial Behavior Questionnaire (STAB), which assesses variation in behaviors that include violence, threats, bullying, theft, and rule-breaking (e.g., littering, vandalism), and has been validated in community, clinical, and adjudicated samples to reliably predict a range of real-world antisocial behaviors [8,10,52]—but not, to date, any behaviors related specifically to disease transmission.

Methods and analysis plans were pre-registered and time-marked April 7th, 2020 at 17:04 EST and all materials, data and code are publicly available (https://osf.io/3429d/).

## Methods

### Participants

A total of 173 participants were recruited from Amazon's Mechanical Turk (MTurk) using a geographical US filter between April 08, 2020 14:25 EST and April 08, 2020 18:18 EST. Participants completed the survey in Qualtrics, which required passing a reCAPTCHA V2. Participants were excluded for being outside the required age range 18–65 (2), reporting that they do not currently live in the U.S. (1), or for failing 2 or more of 3 attention checks (7). In addition, we excluded 32 responses that were flagged as having suspicious location or ISP information using an online tool [53] or which appeared to be duplicates based on human inspection. Results were unchanged when using less strict exclusion criteria for suspicious responses, which is reported in Supplementary Materials (Tables S1-S4 in S1 File). MTurk samples are generally considered to be typical of the general population in terms of most psychological dimensions, though they tend to score higher on negative affect ratings [54], and the need to exclude participants who fail attention checks (as we did) is well documented [55,56].

Participants were compensated $1.00 for completing the survey, which on average lasted 546 ±300 seconds. All study procedures were carried out in accordance with a protocol approved by the Institutional Review Board Committee C at Georgetown University in Washington, DC, and participants provided electronic written informed consent prior to beginning the survey. Demographic characteristics of the final sample are reported in Table 1. Note one participant reported currently working as a healthcare provider.

### Survey

Survey questions related to the COVID-19 pandemic were based upon questions by Anet and colleagues [57], which we expanded upon and adapted for a U.S. sample. Briefly, participants were first asked to respond to the question, "How worried are you about the novel coronavirus (COVID-19) pandemic?" on a 5-point scale from "Not worried at all" to "Very Worried". Participants were then asked about expected impact (i.e. on health or financial status) from COVID-19 and behavioral/ social distancing questions including, "How many times have you

**Table 1. Demographic characteristics of the final sample.**

| Variable | Value |
|---|---|
| Age, M(SD) | 36.3 (10.1) |
| Male/Female (% Male) | 78/53 (59.5%) |
| Race | |
| White | 104 (79.4%) |
| Black | 13 (9.9%) |
| Asian | 6 (4.6%) |
| Other/Mixed | 8 (6.1%) |
| Hispanic/Not (% Hispanic) | 15/116 (11.5%) |
| Education $\geq$ 4-year degree | 74 (56.5%) |
| Employed full or part-time | 112 (85.5%) |
| Household income[a] | |
| $\leq$ $24,999 | 32 (24.6%) |
| $25,000-$89,999 | 77 (59.2%) |
| $\geq$$90,000 | 21 (16.2%) |
| Left house for work[b] | 29/95 (23.4%) |
| PPE use frequency[c] | |
| Never/Rarely | 52 (40.0%) |
| Sometimes/Often/Always | 78 (60.0%) |
| High-risk for serious illness[d] | |
| Self | 29 (23.4%) |
| Lives with someone | 35 (27.8%) |
| Total | 48 (39.0%) |
| Antisocial behavior (STAB-Total) | 54.9 (30.1) |
| Physical Aggression | 17.7 (9.7) |
| Social Aggression | 20.0 (10.1) |
| Rule Breaking | 17.2 (11.1) |

n = 131

[a] One participant did not provide income data

[b] 7 did not respond to whether they had left for work

[c] 1 did not indicate PPE use

[d] 8 did not provide information about risk.

left your home/apartment in the last week?", which participants answered using a drop-down list of values ranging from 0–19 and "More than 20 times" (no participant selected this option).

To assess distance kept from other individuals in the past week, participants were presented with an image of an adult silhouette surrounded by a rectangular border (Fig 1). They were asked to click a point in the image that represents how far away they typically stood from other individuals with the question, "Using the image below, click anywhere to the right of the silhouette that represents how far away you typically stand from other individuals (in the past week)". The selected position was then displayed to the participant, at which point they were able to change their selection if desired prior to advancing. Distance between the silhouette and the selected point was measured in pixels along the x-axis of the image (y-axis information was ignored for statistical analyses). For interpretation, we also converted pixel distance into approximate inches by assuming the silhouette, which depicted a male, represented an average-height man in the US (69.3 inches; 176.0 cm) [58]. Personal protective equipment (PPE)

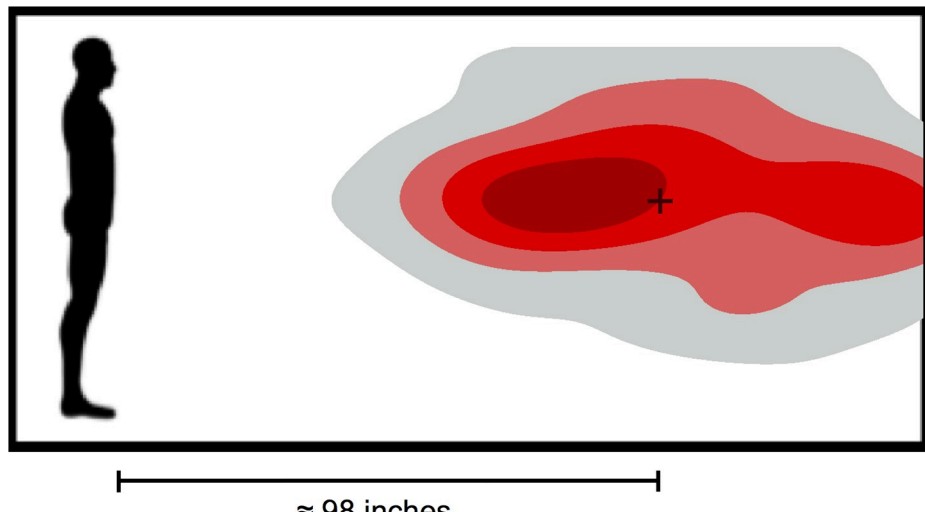

≈ 98 inches

**Fig 1. Distance kept from others in past week.** A bordered image that contained an adult silhouette was used to assess participant-reported distance kept from others. The gray-red heatmap shows how far participants reported standing from other individuals in the past week, with dark maroon indicating a higher density of responses obtained from a kernel density estimation. The mean response coordinate, +, represents a distance of approximately 98 inches (8.2 feet; 2.5 m).

use frequency was determined with the question, "How frequently did you use personal protective equipment such as a mask, face shield and/or gloves when you went outside in the last week?" with response options on a 5-point scale from "Never" = 1 to "Always" = 5.

We inquired about risk for serious illness with the question, "Are you in a high-risk group for becoming seriously ill from COVID-19?" and response choices included "Yes," "No," and "Don't know". We also asked, "Are any of your loved ones in the high-risk group for becoming seriously ill from COVID-19?" and response choices included, "Yes and I live with them", "Yes but I don't live with them", "No", and "Don't know". Subjects were coded as high-risk if they responded "Yes" to the first risk question and/or "Yes and I live with them" to the second risk question.

After completing the COVID-19 specific questions, participants were presented with two optional short-response questions: "Please spend some time thinking about the COVID-19 pandemic and imagining the various things that could happen to you, people you know (such as close friends or relatives), and Americans in general throughout this global pandemic. In a few sentences below, please list your thoughts and feelings about the virus, and include each separate thing that you think could happen" and, "In a few sentences below, please explain why or why not you are practicing social distancing." Participants next completed the STAB questionnaire [10]. Lastly, participants provided demographic and psychological history information. Household income was assessed using the question, "Can you estimate your current household's gross income? This includes all sources of income, including public assistance and social security benefits" and coded responses as follows: 1 = "Under $5,000", 2 = "$5,000–9,999", 3 = "$10,000–14,999", 4 = "$15,000–24,999", 5 = "$25,000–39,999", 6 = "$40,000–59,999", 7 = "$60,000–89,999", 8 = "$90,000–179,999", 9 = "Over $180,000".

## Subtypes of antisocial behavior questionnaire

Participants completed the 32-item STAB questionnaire [10], which inquired about engagement in various antisocial behaviors over the past year using a 5-point scale ("Never", "Hardly ever", "Sometimes", "Frequently", "Nearly all the time"). A summed total antisocial behavior score (STAB-Total) and three subscales were calculated: a 10-item Physical Aggression scale

($\alpha$ = .84-.91), an 11-item Social Aggression scale ($\alpha$ = .83-.90) and an 11-item Rule-Breaking scale ($\alpha$ = .71-.87). The factor structure of the STAB has been previously confirmed in non-clinical adult samples [10].

Note that the STAB includes three items on the Rule-Breaking scale that may have been confounded by economic impacts of COVID-19 ("Had trouble keeping a job", "Failed to pay debts", "Was suspended, expelled, or fired from school or work"). We therefore additionally report results after eliminating these items.

## Statistical inference

Sample size was predetermined with G*Power [59] using the main outcome variable of number of times left home in the past week. We anticipated a base rate of $\beta 0$ = .286 (2 times in 7 days), a 25% increase ($\beta 1$ = 1.25) associated with antisocial behavior, and a moderate association between covariates and the main predictor ($R^2$ other x = .25). Setting alpha = .05 and power = .90, we calculated that a sample size of 138 would provide sufficient power. We expected this sample size to also provide sufficient power for the distance and worry multiple linear regression analyses estimating effect size as Cohen's $f^2$ = 0.1, n predictors = 5, alpha = .05 and power = .90 (critical n = 108). Assuming the need to exclude 20% of responses, we collected n = 173. We excluded more responses than anticipated (approximately 30%) leaving us with n = 131 rather than the expected 138; however, our final sample size provided greater than standard power (.80) for all analyses (times left home post-hoc power = .85; distance post-hoc power = .92, worry post-hoc power = .94). All following statistical analyses were completed in Stata 15 (StataCorp. 2017. College Station, TX).

## Results

### Times left home in past week

Participants reported leaving their home a median of 2 times in the past week (IQR = 1–3; Min = 0; Max = 15). To investigate whether leaving the home more frequently is associated with general antisocial tendencies, we applied statistical count models with the dependent variable set as times left in the past week and STAB-Total as an independent predictor. Models included the following covariates: age, sex, education, household income, whether the participant left home for work in the last week, and whether the participant was at high risk or lives with someone at high risk. Variables were entered in steps with basic demographics entered in Step 1, COVID-19-related covariates entered in Step 2, and STAB-Total entered in Step 3. A goodness-of-fit test from a Poisson model indicated the data for the full model were over-dispersed (deviance g.o.f. = 220.17, p < .001), we therefore applied a negative binomial model; the likelihood-ratio test of the resulting alpha distribution parameter was greater than 0, indicating that the negative binomial model provided a better fit than Poisson ($\alpha$ = .37, 95% CI = [.21, .64], $\chi^2$ = 34.60, p < .001). In the model, antisocial behavior scores and age are mean-centered, sex is coded as 0 = male and 1 = female, education is coded as 0 = <4-year degree and 1 = ≥4-year degree, household income was entered as a mean-centered continuous variable (see Methods for coding), left house for work in the past week is coded as 0 = no and 1 = yes, high-risk for serious illness (for self or someone the participant lives with) is coded as 0 = no and 1 = yes. Cases were excluded list-wise for missing data (household income, 1; left house for work, 7; high-risk, 8).

Results showed that frequency of leaving home during the COVID-19 pandemic was associated with overall antisocial tendencies, such that one standard deviation increase in STAB-Total was associated with 21.5% more incidents of leaving after adjusting for covariates (Table 2). Fig 2A displays this relationship while holding other covariates at their mean. As expected, leaving home for work in the past week was associated with 65.0% more incidents.

**Table 2. Negative binomial regression predicting the number of times participants left their home in the past week.**

| | Step 1 | | | Step 2 | | | Step 3 | | |
|---|---|---|---|---|---|---|---|---|---|
| | IRR | 95% CI | p | IRR | 95% CI | p | IRR | 95% CI | p |
| Constant | 2.589 | 1.888, 3.550 | | 2.207 | 1.587, 3.069 | | 2.578 | 1.818, 3.657 | |
| Age | 1.011 | 0.994, 1.029 | .195 | 1.013 | 0.997, 1.030 | .111 | 1.013 | 0.997, 1.030 | .105 |
| Sex | 0.816 | 0.561, 1.189 | .290 | 0.924 | 0.642, 1.330 | .671 | 0.925 | 0.647, 1.322 | .669 |
| Education | 0.996 | 0.682, 1.456 | .985 | 0.848 | 0.584, 1.231 | .385 | 0.725 | 0.491, 1.071 | .106 |
| Household income | 0.937 | 0.838, 1.048 | .252 | 0.958 | 0.861, 1.066 | .432 | 1.004 | 0.897, 1.123 | .946 |
| Left for work | | | | 2.029*** | 1.376, 2.991 | < .001 | 1.650* | 1.084, 2.510 | .019 |
| High-risk | | | | 0.988 | 0.699, 1.396 | .945 | 0.894 | 0.630, 1.270 | .531 |
| Antisocial behavior | | | | | | | 1.007* | 1.001, 1.013 | .024 |

n = 116. Step 1: $\chi^2(4) = 3.74$, p = .443. Step 2: $\chi^2(6) = 16.00$, p = .014; $\Delta\chi^2 = 12.83$, p = .002. Step 3: $\chi^2(7) = 21.00$, p = .004; $\Delta\chi^2 = 5.09$, p = .024.

*p < .05

**p < .01

***p < .001.

Originally pre-registered analyses did not include education and income covariates, which were added based on subsequent feedback. However, we report all pre-registered statistical models fully in the supplemental materials, which show no qualitative differences in results (Tables S5-S7 in S1 File).

## Distance kept from others in past week

We next assessed whether antisocial tendencies were associated with estimates of real-world social distancing. Distance was measured as the horizontal distance (x-axis only) between the silhouette image and the selected typical standing distance in pixels (M = 392.6, SD = 130.2, 95% CI = [369.2, 416.1], Fig 1). Approximate conversion to inches indicated that on average participants reported standing 98.2 inches (8.2 feet; 2.5 m) away from others (M = 98.2, SD = 32.6, 95% CI = [92.4, 104.1]). We applied a multiple linear regression predicting distance in pixels from total antisociality score while including the following covariates: age, sex, education, household income, whether the participant was at high-risk or lives with someone at high-risk, and PPE use frequency. PPE use frequency was entered as a continuous variable. Variables were again entered in steps; with basic demographics entered in Step 1, COVID-19-related covariates entered in Step 2, and STAB-Total entered in Step 3. Cases were excluded list-wise for missing or invalid (e.g. participant's response was on the silhouette) responses (household income, 1; distance from silhouette, 10; PPE use frequency, 1; high-risk, 8).

As expected, antisociality was associated with reduced reported distance kept from others, such that one standard deviation increase in STAB-Total was associated with 52.8 fewer pixels (i.e. 13.2 inches; 33.6 cm) of social distance after adjusting for covariates (Table 3). Fig 2B displays this relationship while holding other covariates at their mean. Increased PPE use frequency was associated with increased reported distance kept from others while having at least a 4-year college degree was associated with reduced distance kept from others.

## Assessing possible confounds related to economic or financial impact of COVID-19

Three items on the STAB questionnaire could relate to economic or financial impacts of COVID-19 (e.g. job loss, inability to pay debts). Therefore, we conducted the same analyses after removing these three items to confirm the relationships between antisocial behavior and

reduced social distancing persisted. In these post-hoc tests, results were qualitatively unchanged and full tables are reported in Supplemental Materials (Tables S8-S9 in **S1 File**; a modified correlation table is also reported in Table S10 in **S1 File**). The modified STAB-Total remained a significant predictor for times leaving the home (IRR = 1.008, 95% CI = [1.001, 1.015], p = .028) and distance kept from others (B = -1.928, 95% CI = [-2.964, -0.892], p < .001), indicating that potential economic impacts of COVID-19 did not underpin the observed relationships between antisociality and social distancing.

Finally, following observations that the economic impacts of COVID-19 disproportionately affect individuals from minority groups, we observed that Black and Hispanic participants more frequently reported needing to leave home for work during the early, stay-at-home phase of the pandemic (58% relative to 14% of non-Black or Hispanic subjects, $\chi^2$ = 21.6, p < .001).

## Reported worry about COVID-19

Participants on average reported moderate-to-high levels of worry about the coronavirus with a mean response of 3.63 on the 5-point scale (M = 3.63, SD = 1.10, 95% CI = [3.44, 3.82]). We next considered the hypothesis that antisociality is associated with reduced worry about COVID-19. We thus applied a multiple linear regression predicting the dependent worry variable from STAB-Total and included the following covariates: age, sex, education and whether the participant was at high risk or lives with someone at high risk. In this model (F(5,117) = 2.51, $R^2$ = .097, p = .034, n = 123), antisocial behavior score was not a significant predictor of worry (B = 0.005, 95% CI = [-0.002, 0.012], p = .161). The only significant predictor in the model was the high-risk variable, which was associated with increased worry (B = 0.551, 95% CI = [0.145, 0.958], p = .008). We assessed the same variables using an ordered logit model and observed the same results.

## Correlations among variables

Inter-correlations between study variables are reported in Table 4. We applied Spearman's rank correlations due to the non-normality or count/ordinal scale of most variables. We

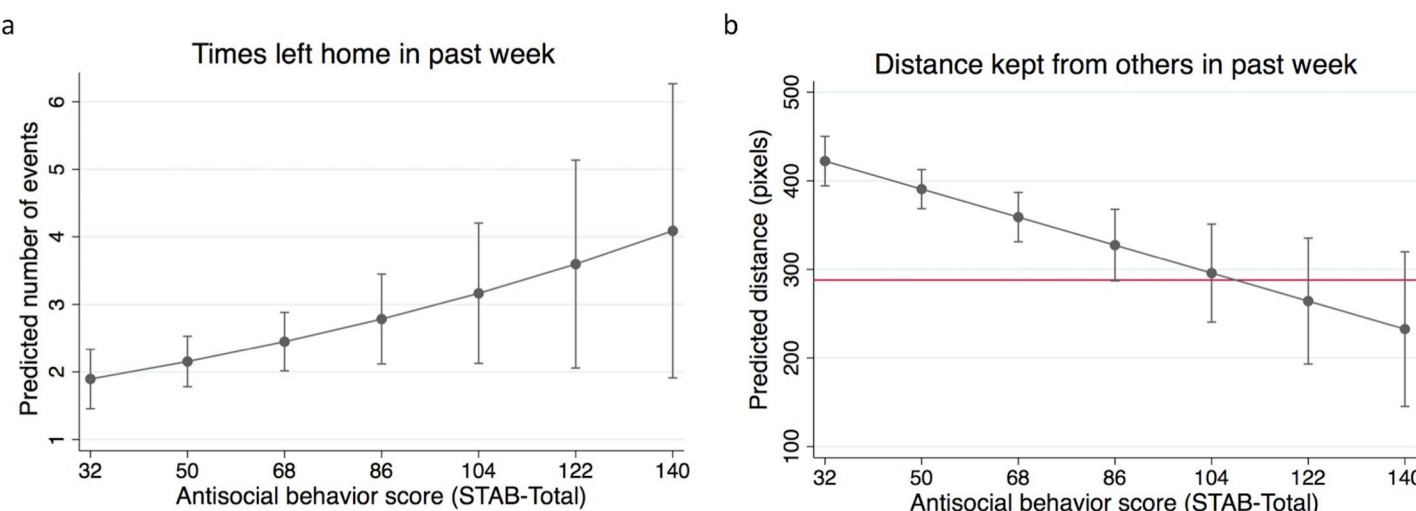

**Fig 2. Antisociality is associated with leaving the home more frequently and standing closer to others during the COVID-19 pandemic. (a)** Adjusted predictions from the negative binomial regression indicate that leaving the home more frequently is modestly positively associated with antisociality. **(b)** Adjusted predictions of distance kept from others outside of the home in the past week is negatively associated with antisociality; the red line denotes the government recommended distance of 6 feet (1.8 m). Error bars represent 95% CI of the mean.

**Table 3. Multiple linear regression predicting distance in pixels.**

| | Step 1 | | | Step 2 | | | Step 3 | | |
|---|---|---|---|---|---|---|---|---|---|
| | **B** | **95% CI** | **p** | **B** | **95% CI** | **P** | **B** | **95% CI** | **p** |
| Constant | 417.414 | 375.623, 459.205 | | 379.318 | 319.687, 438.948 | | 346.500 | 287.482, 405.519 | |
| Age | -0.239 | -2.583, 2.105 | .840 | -0.187 | -2.541, 2.168 | .875 | -0.264 | -2.489, 1.961 | .815 |
| Sex | 46.467 | -2.016, 94.950 | .060 | 44.173 | -4.242, 92.588 | .073 | 31.109 | -15.171, 77.389 | .186 |
| Education | -85.229** | -137.226, -33.233 | .002 | -86.031** | -137.771, -34.291 | .001 | -66.380** | -116.389, -16.372 | .010 |
| Household Income | 15.485* | 1.028, 29.943 | .036 | 15.542* | 1.164, 29.920 | .034 | 10.663 | -3.171, 24.497 | .129 |
| High-risk | | | | 21.907 | -27.502, 71.316 | .381 | 33.215 | -13.863, 80.293 | .165 |
| PPE use frequency | | | | 10.924 | -3.666, 25.513 | .141 | 16.089* | 2.029, 30.150 | .025 |
| Antisocial behavior | | | | | | | -1.756*** | -2.695, -0.817 | < .001 |

n = 113; 1 pixel is approximately equivalent to 0.25 inches (0.64 cm). Step 1: $F_{(4,108)}$ = 3.81, $R^2$ = .091, p = .006. Step 2: $F_{(6,106)}$ = 3.12, $R^2$ = .150, p = .008; $\Delta R^2$ = .026, p = .200. Step 3: $F_{(7,105)}$ = 4.95, $R^2$ = .248, p < .001; $\Delta R^2$ = .098, p < .001. B represents unstandardized beta coefficients.

*p < .05

**p < .01

***p < .001.

observed a significant relationship between the two main dependent variables such that leaving the home more frequently was associated with reduced distance kept from others (rho = -.38, p < .001), suggesting a consistent violation of social distancing norms in some individuals. The bivariate correlation between STAB-Total and frequency of leaving the house was statistically significant (rho = .27, p = .005), whereas the correlation with reported distance kept reached trend level (rho = -.18, p = .062). Level of worry revealed a positive relationship with PPE use frequency (rho = .25, p = .010) and high-risk status (rho = .31, p = .001), but had no significant relationship with other variables. Subscales of the STAB questionnaire were highly intercorrelated (rho values ranged from .65 to .85), which supported our use of a total score.

**Table 4. Intercorrelations among variables.**

| | 1 | 2 | 3 | 4 | 5 | 6 | 7 | 8 | 9 | 10 | 11 | 12 | 13 | 14 |
|---|---|---|---|---|---|---|---|---|---|---|---|---|---|---|
| 1. Age | - | | | | | | | | | | | | | |
| 2. Sex | .12 | - | | | | | | | | | | | | |
| 3. Education | .13 | .05 | - | | | | | | | | | | | |
| 4. Income | .02 | .10 | .38*** | - | | | | | | | | | | |
| 5. Left for Work | -.16 | -.20* | .10 | .07 | - | | | | | | | | | |
| 6. High-Risk | .04 | .11 | .00 | .02 | -.01 | - | | | | | | | | |
| 7. PPE use frequency | -.09 | .02 | .02 | .03 | -.07 | .07 | - | | | | | | | |
| 8. Times left house | -.02 | -.12 | .04 | -.03 | .34*** | -.05 | -.11 | - | | | | | | |
| 9. Distance kept | -.05 | .16 | -.21* | .07 | -.32** | .16 | .15 | -.38*** | - | | | | | |
| 10. Worry about COVID-19 | .05 | .14 | -.01 | -.07 | -.10 | .31** | .25** | -.17 | .16 | - | | | | |
| 11. Physical Aggression | -.13 | -.12 | .04 | -.13 | .20* | .16 | .07 | .25** | -.17 | .15 | - | | | |
| 12. Social Aggression | -.03 | -.07 | .04 | -.10 | .24* | .10 | .07 | .25** | -.20* | .12 | .85*** | - | | |
| 13. Rule Breaking | -.18 | -.20* | .03 | -.22* | .22* | .10 | .13 | .34*** | -.21* | .13 | .71*** | .65*** | - | |
| 14. STAB-Total | -.09 | -.09 | .01 | -.13 | .21* | .14 | .09 | .27** | -.18 | .14 | .95*** | .96*** | .74*** | - |

All correlations are Spearman rho values; n = 107.

*p < .05

**p < .01

***p < .05 Bonferroni corrected for 91 comparisons.

## Discussion

We find that antisociality is associated with reduced social distancing during the COVID-19 pandemic—specifically in early April of 2020, a period of high uncertainty, awareness of community disease spread in the U.S., government ordered stay-at-home guidelines, and news of well-known public figures being treated in intensive care units. In line with our pre-registered hypotheses, we observed that antisociality was associated with leaving the home more frequently and standing physically closer to others, even after controlling for demographic and education variables, risk for serious illness, leaving the home for work, and use of protective equipment. These findings are the first to link behaviors that increase the risk of coronavirus disease-transmission to antisocial behavior more generally, reinforcing the importance of understanding variation in antisocial tendencies for public health.

An estimated 24.4% of our sample reported violating social distancing norms in early April. This includes the 8.4% who reported leaving the home more than five times in the past week despite not leaving for work, and the 19.8% who reported standing less than 6 feet from other individuals while outside. Of these individuals, 9.4% (3/32) also reported having "flu-like" symptoms, which is a small but potentially meaningful sample. When queried about their rationale for violating social distance recommendations using an open-ended question format, responses from the 32 violators primarily referenced self-oriented concerns, such as, "*I don't think much will happen to me personally, other than not being able to buy groceries whenever I want. I just hope it slows down/ends soon,*" "*I worry that my retirement accounts won't recover,*" and "*I certainly cannot move ahead in life* [sic] *to the economy gets going again and the restrictions are lifted.*" The proportion of respondents who reported violating social distance norms is consistent with the observation that the transmission of infectious agents such as the novel coronavirus follows a 20/80 rule, meaning that 80% of cases arise from only 20% of the infected population [28,29], and that a few "super-spreaders" disproportionally infect large numbers of people. Both physiological (e.g. viral shedding) and behavioral (e.g. contact length and frequency) factors are considered important features of "super-spreaders" [60] and through this report, we hope to convey that antisociality may serve as a significant, albeit modest, contributing factor for behaviors relevant to infectious disease spread. Our results, if confirmed and extended, may suggest that even a small population of antisocial hosts could have important implications for the propagation of a global pandemic like COVID-19.

Our findings align with results from a number of published and unpublished papers that report on psychological correlates of behaviors and attitudes toward COVID-19. A large sample of 22-year-olds enrolled from a Swiss longitudinal study showed that previous engagement in delinquent behaviors was associated with reduced social distancing and worse hygiene behaviors related to the virus [61], suggesting that our results similarly apply among a young adult-limited sample, which represents a peak age of antisocial behavior [62]. An online sample of 502 adult participants conducted in late March, found that psychopathic traits (which are risk factors for persistent antisocial behavior) were associated with reduced social distancing and worse hygiene—and even with the intent to knowingly expose others to risk and reduced appeal of a compassionate public-health message [63], indicating the challenges inherent in trying to sway highly antisocial populations' behaviors toward public-health norms. These observations align with findings of typically reduced preferred social distances in high-psychopathy samples [64]. Nowak and colleagues similarly report that psychopathic traits in a Polish sample (n = 755) were associated with increased stockpiling of supplies (e.g. food, masks, sanitizer) and decreased engagement in disease spread prevention behaviors in March 2020, and furthermore discuss the mediating effects of various health beliefs [65]. Recently published work has linked other personality traits to COVID-19-related behaviors, for

example, finding emotionality and conscientiousness associated with increased toilet paper stockpiling [66] and relating conspiracy theory beliefs to COVID-19 attitudes about government responses [67]. Our findings also add the important information that, contrary to our hypothesis, antisociality was unrelated to worry about COVID-19, despite antisocial behaviors often being associated with reduced fear or anxiety [68–73]. This raises questions about whether fear-based approaches to shifting the behavior of antisocial individuals would be effective.

We observed that subjects who reported leaving the home for any reason more frequently also reported leaving home for work in the past week. Critically, our results show that antisociality remains a significant independent predictor of leaving the home more frequently even after accounting for leaving home to work. Thus, antisociality appears to contribute to the disregard for health guidelines in some subjects, while independently the need to leave for work contributes to the frequency of leaving home for others. These findings should not be interpreted as suggesting that leaving home to work reflects antisociality. Participants who reported leaving home to work may have included essential workers performing vital tasks for the community (note these workers disproportionately included Black and Hispanic individuals, consistent with work in large representative samples recognizing economic and health disparities during the COVID-19 pandemic including increased exposure risk related to employment [74–76]). In post-hoc analyses, we show that removing economic items from the antisocial behavior measure (e.g. failed to pay debts, had trouble keeping a job) had no effect on the results. This, in combination with the inclusion of socioeconomic covariates in our regression models, provides evidence that the relationship between antisociality and reduced social distancing persists when adjusting for socioeconomic factors that independently contributed to reduced social distancing.

The effect size of antisociality on leaving the home more frequently was modest—a one standard deviation increase in antisociality score was associated with 21.5% more events (subjects who reported leaving home for work had 65.0% more events). In the model predicting distance kept from others, a one standard deviation increase in antisociality was associated with keeping 13.2 inches (33.6 cm) less distance from others. Higher education level and lower PPE use frequency were also associated with reduced distance kept from others. Since only one subject reported working as a healthcare provider, this unexpected effect of higher educational status predicting reduced distance does not appear to be driven by the fact that essential healthcare professionals may need to be physically close to patients. The result could be related to geographical or population density differences between subjects with varying levels of education. Higher PPE use frequency was associated with increased distance kept from others, as might be expected by more cautious individuals given that PPE use frequency was also correlated with increased worry about COVID-19. However, at the time of data collection in early April 2020, face masks and other PPE were expensive and scarce in the U.S. and public messaging was inconsistent [34–36,77]. Throughout the preceding months, U.S. government officials urged citizens not to buy face masks, but reversed this messaging on April 3rd to recommend public use of face masks [35]. Due to these recommendation inconsistencies, unequal access to PPE in the U.S. at the time of data collection, and other research linking psychopathic traits to stockpiling of PPE [65], our pre-registered analyses focused on the social distancing guidelines and stay-at-home orders that were well established [78] at the time and affected 94% of the U.S. population [31,32]. Especially because 40.0% of respondents reported never or rarely using PPE in the past week, we suggest caution when drawing conclusions from this variable, as it may simply reflect access to PPE at this particular time point.

A few additional limitations should be considered when interpreting our results. While our sample roughly approximates major demographics of U.S. adults 18–65 (76% White, 14%

Black, 7% Asian; 23% Hispanic; 50% female; national data on adults 18–64 as of 2019 [79]) and includes participants from 38 states, it is not nationally representative due to our recruitment using Amazon's MTurk rather than via a sampling panel and due to the relatively small— though sufficiently powered—sample size. MTurk samples are also not representative of U.S. sociodemographic variables and tend to include individuals with higher education and income [80]. Replication in a larger representative U.S. sample will be important. The study also used a single time point for data collection and relied on retrospective report of social distancing behaviors in the past week, which may be subject to bias. This design did not permit us to track disease status or spread in relation to antisociality (note only one participant reported a diagnosis of COVID-19 at the time of data collection). Multicollinearity among subscores of the STAB measure made it impractical to test for predictive differences between subtypes of antisocial behavior, though correlations indicate the rule-breaking subscore is most strongly associated with violating social distancing guidelines (we originally hypothesized social aggression). It is likely that other variables not measured in the present study could also account for variations in social distancing. Personality traits such as extraversion, agreeableness, or honesty-humility might be related to aspects of social distancing behavior, as well as to antisociality. For example, extraverted personality may be associated with antisociality, social distancing, and having a job that requires leaving the home. Were future work to identify the contribution of such factors to distancing behavior, it could have important implications for interpreting our results and for targeting or refining future public health approaches to a pandemic. Recent research also indicates a relationship between political beliefs and social distancing during the pandemic [81]; however, our study did not measure any political variables and is unable to test related questions.

Additional limitations include that worry about COVID-19 was measured on a 5-point scale, which may not have provided sufficiently detailed information to detect a relationship with antisociality. Our main outcome measures of distances were novel. Although their strong interrelationship supports their validity, additional assessment of these measures are needed. For example, our survey did not distinguish between reasons for leaving the home (e.g. grocery shopping versus outdoor exercise), which may provide useful information for future research. The silhouette image had no specific contextualization and could be adapted in the future to include, for instance, the backdrop of a grocery store, which may increase measurement precision. With these limitations in mind, we note a strength of this study was that all hypotheses and methods were pre-registered and data are publicly available.

In conclusion, we found that a validated measure of antisociality helps explain reduced social distancing during the COVID-19 pandemic, suggesting antisociality is important to consider as a behavioral factor for communicable disease spread. These findings warrant future research to test manipulations aimed at boosting adherence to social distancing guidelines specifically when considering antisociality or related factors, for example by evaluating prompts signaling empathic versus self-oriented motivational cues.

## Supporting information

**S1 File.**
(PDF)

## Acknowledgments

We thank Hannah Savitz for project assistance and our participants for their time and effort.

## Author Contributions

**Conceptualization:** Katherine O'Connell, Kathryn Berluti, Shawn A. Rhoads, Abigail A. Marsh.

**Formal analysis:** Katherine O'Connell, Kathryn Berluti.

**Investigation:** Katherine O'Connell, Shawn A. Rhoads.

**Supervision:** Abigail A. Marsh.

**Visualization:** Katherine O'Connell.

**Writing – original draft:** Katherine O'Connell.

**Writing – review & editing:** Kathryn Berluti, Shawn A. Rhoads, Abigail A. Marsh.

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
