## [Decision Letter · Decision Letter 0]

14 Aug 2020

PONE-D-20-19373

Reduced social distancing during the COVID-19 pandemic is associated with antisocial behaviors in an online United States sample

PLOS ONE

Dear Dr. O'Connell,

I am writing with a decision on your manuscript “Reduced social distancing during the COVID-19 pandemic is associated with antisocial behaviors in an online United States sample.” I have received reviews back from three scholars with expertise with the methods and topic of your study. I reviewed the paper independently. The reviewers and I agree that the paper is well-written, concise and clear, and represents timely results relevant to our current public health crisis. There were also various concerns and limitations noted, some that involve potential confounds that need to be addressed before the paper is ready for publication. As such, I will recommend that the authors revise the manuscript and address all of the reviewer comments for resubmission to the journal. At that point, the paper will go through another round of reviews.

The reviewers provided very clear comments and recommendations, and there is a lot of consistency across reviewers. As you can see, both Reviewers 1 and 3 believe that there are several potential confounds that were not assessed or considered, including employment anxiety or fear of loss of employment, type of work and essential worker status, economic concerns, income and/or political ideology. Reviewer 1’s alternative explanations are compelling (except for the point about PPE use and social distancing, which the reviewer interpreted incorrectly), including regarding potential confounds written into the Rule Breaking subscale. Reviewer 2 noted relevant covariates that were mentioned in the registered report but not found in the manuscript (e.g., occupation). The question is whether you have the data or other possible ways to rule out these confounds or at least convey why the current analyses are sufficient and impactful as they are. The authors should also address comments about how antisocial behavior was measured, some of the potentially confounding items, and the relevance of separate facets of the construct. There is also a sense that some of the conclusions are overstated, given the data, and more tentative language should be used that does not overattribute effects to antisocial tendencies. In addition, the effect sizes are modest, especially in the first analysis, and that could be addressed better. Finally, I agree with the reviewers that PPE use would be an ideal outcome measure, and it’s unclear why it was included as a covariate instead. The (often ideologically-driven) controversies around PPE use likely play an important role in the continued spread of the virus.

The reviewers included many other recommendations that I think will be useful as you revise the paper for resubmission. Please attach a cover letter describing how you addressed all comments with your revised manuscript. Thank you for submitting your work for consideration in PLOS ONE.

We look forward to receiving your revised manuscript.

Kind regards,

Edelyn Verona

Academic Editor

PLOS ONE

Journal Requirements:

Reviewers' comments:

Reviewer's Responses to Questions

**Comments to the Author**

1. Is the manuscript technically sound, and do the data support the conclusions?

Reviewer #1: No

Reviewer #2: Partly

Reviewer #3: Partly

2. Has the statistical analysis been performed appropriately and rigorously? 

Reviewer #1: Yes

Reviewer #2: Yes

Reviewer #3: Yes

3. Have the authors made all data underlying the findings in their manuscript fully available?

Reviewer #1: Yes

Reviewer #2: Yes

Reviewer #3: Yes

4. Is the manuscript presented in an intelligible fashion and written in standard English?

Reviewer #1: Yes

Reviewer #2: Yes

Reviewer #3: Yes

5. Review Comments to the Author

Reviewer #1: The current study presents an analysis of 117 participants who completed self-report questionnaires regarding their COVID-19 pandemic-related compliance with social distancing and use of personal protective equipment (PPE, “mask, face shield and/or gloves”) in the previous week as well as basic demographic questions and the 32 item Antisocial Behavior Questionnaire (STAB) inquiring about antisocial behaviors across physical and social aggression, as well as rule breaking domains in the past year.

The authors concluded based on regression analyses including totals scores on the STAB and demographic variables that “antisociality predicted reduced social distancing during the COVID-19 pandemic.” The study has some strengths, including an interesting, novel approach to assess subjective social distancing and the authors should be commended for attempting to study of interesting variable in a very timely topic.

I have various concerns about the validity of the study’s conclusions that I outline below from major to minor below.

1) An examination of table 2 shows that the predictors for participants leaving their home in past week were “left for work” and “antisocial behavior” (i.e., STAB total score). The fact that “left for work” is as strong a predictor as “antisocial behavior” for leaving home was unaddressed by the authors and ignores a major alternative explanation for results of study: Economics. The authors may argue that because STAB total scores predicted leaving home in last week independently from “left for work” their conclusions stand. However, an examination of Table 4 indicates that “Rule-Breaking” subscale from the STAB was the most strongly related to “Times Left the House” and the only one to relate to “Distance Kept.” An examination of the items of the “Rule Breaking” scale from the STAB shows that there are three items that are all labor-related and could be endorsed by many people in their sample due to pandemic associate massive layoffs (”was suspended, expelled, or fired from school or work”; “had trouble keeping a job”; “failed to pay debts”). It would be interesting to see the rates of endorsement of items in the “Rule Breaking” subscale and see if after removing these 3 items from analyses the results would stand. It is very possible that the observed results for the “leaving home” outcome are driven by people who must leave for work because they are “essential” either in the higher end of education (e.g., medical personnel) but also in the lower end (cashiers, truckers, agricultural workers) who are also the most vulnerable to losing their jobs in the past year and thus endorse those 3 “Rule Breaking” items that basically penalize economic vicissitudes.

2) My hypothesis above is supported by the authors’ own data. Table 3 shows that less self- reported distance from others is predicted by higher education and more PPE use. In short, possibly by people who are essential workers in higher and lower SES strata who by the nature of their work (physician, cashier) have to leave home AND keep less distance from others. Again, the fact that higher education and PPE use predicts lower distance is glossed over and not addressed respectively in the discussion of the article, and supports the possibility that what we are seeing is the grim reality of the pandemic: Essential workers are having to put themselves at higher risk and the way they cope with is through increased PPE use. Those who have to leave home and be closer to others may do it because they have been “fired from work” and thus “had trouble keeping a job” and consequently are have “failed to pay debts.” Certainly not antisociality.

3) My hypothesis is further supported also by the authors’ own report that none of the STAB subscales or total score were related to PPE use. This further suggests that the reported relationship between “antisociality” and pandemic social distancing norms that the authors report have little to do with flaunting of social norms (mask wearing being one of the most contentious pandemic norms in the U.S.) and more with financial need to go to work. If antisociality assessed by the STAB which includes hitting, threatening, and swearing indeed was affecting these norms, some relation to PPE (arguably an easier to break norm) would be observed but that is not the case.

4) Incidentally, the data suggests that the contentiousness of social distancing and mask-wearing (or not) in the U.S. breaks along party/ideological identification with more democrat/liberal individuals endorsing more mask wearing whereas more republican/conservatives endorse less mask wearing (Pew Report, 2020). I don’t know if the authors asked any data in this respect but in general, republican/conservative individuals tend to endorse more traditional, norm following attitudes. Those data would militate against “antisocial behavior” as an explanation for failure to maintain social distance or remaining at home. The possibility that blue-collar jobs that are being threatened in this economy may be held by persons with more conservative ideology could however be playing a role. Or that more conservative leaders are disavowing social distancing and mask wearing citing economic reasons. Indeed, some of the qualitative data mentioned by the autors in p.18 suggest that economic anxiety is at the core of some of their participants’ behavior (“I worry that my retirement accounts won’t recover” “…cannot move ahead in life [sic]to the economy gets going again…”If the authors had a variable to address the influence of political ideology on COVID-19 related health norms would strengthen their study significantly. Alternatively, (although I am often reluctant to be the reviewer who says ‘collect more data’) doing another rapid survey on MTurk as the authors did, with a larger sample that assesses political ideology, recent loss of job because of the pandemic and economic stress due to it, and controls for it in their models could provide far more illuminating results.

5) An important variable related to the above concerns also not addressed in the analysis is the effect of ethnicity in their findings. The pandemic has hit lower SES workers and in especially Black and Latino workers and communities. The numbers of individuals from these communities are underrepresented in the study’s sample (perhaps another reason to continue collecting data and oversampling these populations) but nonetheless, it would behoove the authors to explore the interaction between these two ethnicities and endorsement of the three labor related items from the “Rule Breaking” scale of the STAB. The results may show yet another grim reality from the pandemic: The endorsement of having to leave home more often and be closer to other individuals is not due to “antisociality” but due to Black and Latino households having a higher rates of job losses during the pandemic, higher rates of burden for the caring of more family members, less work flexibility, and increased need to find more dangerous jobs during a pandemic.

6) Given the above concerns, the authors should explain why they chose to use the total score of the STAB-Total score in the regression analyses given that most of the variance is accounted for by the “rule breaking” scale. As I mentioned, I suspect that their results are better explained by endorsement of the 3 labor-related items in it and related to pandemic related massive layoffs rather than frank rule breaking. However, even if after removing these items, the (relatively) weak effect results hold, the authors would have to find a way to better contextualize their results. Otherwise they could be summarized as “people who do not follow strong social norms of behavior like stealing, and destroying private and public property are more likely to leave the home (paradoxically, possibly to work) and keep slightly less distance from others during a pandemic.” To be fair, the authors should be commended for trying to study such a fluid and novel phenomenon. However, the sample size, measures and thoroughness of analyses compares unfavorably with other efforts that are emerging with similar topic (e.g., Georgiou et al., 2020; Nowak et al., 2020).

Thank you for the opportunity to review this interesting manuscript.

References

Bartłomiej Nowak, Paweł Brzóska, Jarosław Piotrowski, Constantine Sedikides, Magdalena Żemojtel-Piotrowska, Peter K. Jonason, Adaptive and maladaptive behavior during the COVID-19 pandemic: The roles of Dark Triad traits, collective narcissism, and health beliefs, Personality and Individual Differences, Volume 167, 2020, 110232, ISSN 0191-8869, https://doi.org/10.1016/j.paid.2020.110232.

Neophytos Georgiou, Paul Delfabbro, Ryan Balzan, COVID-19-related conspiracy beliefs and their relationship with perceived stress and pre-existing conspiracy beliefs, Personality and Individual Differences, Volume 166, 2020, 110201, ISSN 0191-8869, https://doi.org/10.1016/j.paid.2020.110201.

“Republicans, Democrats Move Even Further Apart in Coronavirus Concerns.” Pew Research Center - U.S. Politics & Policy, Pew Research Center, 24 July 2020.

www.pewresearch.org/politics/2020/06/25/republicans-democrats-move-even-further-apart-in-coronavirus-concerns/.

Reviewer #2: This report uses data collected from 131 MTurk workers to evaluate the associations between antisociality and three variables related to feelings and behaviors during the Covid-19 pandemic in the Spring of 2020: the number of times participants left their homes during the past week, an estimation of typical social distancing, and worry about the pandemic.

Reviewer background: I read the paper and the pre-registration. I did not check the analyses myself as I do not use STATA.

Strengths: The topic is interesting and timely. The work was pre-registered. The report was efficiently written. Count regression models were used for the count DV. I think the main result is interesting in that it suggested (at least to me) an association between general antisociality and certain ways of acting in a pandemic that are harmful to others. The social distancing measure was novel.

Weaknesses: Cross-sectional data, sample might be selective, some deviations from the pre-registration. There was a power analysis but as an individual difference researcher, I typically prefer larger samples for more precise estimation of relevant coefficients.

Three Pre-Registration Comments.

1. The authors pre-registered the following covariates for the models in Tables 2 & 3 = age, sex, use of PPE, high risk group, and occupation requiring one to leave home [omitted for the distance DV]. Tables 2 & 3 also include education. So, education was not pre-registered (as best as I could tell). This should be acknowledged assuming I am not missing something.

2. The pre-registration also notes a sub-aim to test whether the STAB “effects” are primarily driven by the social aggression subscale. This issue was likely hard to test given that the STAB scales are strongly correlated but those null results could be summarized in the paper with a sentence. (If anything, it looks like the Rule Breaking scale had the largest correlations, but I doubt there would be evidence that that the coefficients differed from one another).

3. It should probably be noted that the authors wanted to have a sample of 138 but had to deal with 131 for the first analysis. This is not a deal-breaker at all to me, but I just like to see deviations noted in papers that have pre-registration. (The authors did something savvy as well by setting power to .90 rather than the typical .80 value that often makes little sense.)

General Comments

4. The zero-order STAB effect for distancing was not p < .05 best as I could tell. That might be worth a comment.

5. I think adding means and SDs to Table 4 would be useful. I would also add all the variables used in the regression models to this table. For example, left for work and high-risk could be included.

6. I think the attention checks for MTurk workers were savvy. The authors might want to say more about the use of MTurk workers. I assume that the state data was too sparse to try a multilevel model given that states had different policies.

7. The authors could add a few sentences to describe the pandemic conditions in the USA during April 8 and the preceding week.

8. The high-risk variable could be described a bit more in the paper.

9. The power analysis used a poisson regression but the negative binomial was reported. The data were likely more over dispersed than ideal for the standard poisson model so the negative binomial is more appropriate and I think they have better type I error control in this case. (I am working from home and my texts are in my office so I might be misremembering these details). Anyways, I don’t know what (if anything) this difference in models means for the power analysis.

10. The reported worry single item DV might have more measurement error than ideal for testing the hypothesis. This could be noted as a limitation.

11. I think the conclusions on the top of page 19 (esp. the “even a small population of antisocial hosts…”) were a bit too strongly worded for my taste.

12. I think the Discussion was otherwise interesting and I appreciated the connection to other recent studies.

Reviewer #3: This manuscript examines whether antisocial behavior is associated with reduced social distancing during the COVID-19 pandemic. This paper addresses a timely question and does a good job of explaining the relevance of the work to public health. However, there are several issues with the conceptualization of the study and the methods/results. These are described below.

Background Section:

• The authors discuss antisociality as an underlying phenotype. The citations provided do not provide sufficient support for this argument. The following is an example of an article that would provide much better support for this assertion:

Niv, S., Tuvblad, C., Raine, A., & Baker, L. A. (2013). Aggression and rule-breaking: Heratibility and stability of antisocial behavior problems in childhood and adolescence. Journal of Criminal Justice, 41.

• Generally, antisocial behavior is discussed as a unitary construct. This is problematic, as even one of the articles cited by the authors argues that there is only moderate overlap in genetic influences on aggression and rule-breaking behavior (Burt, 2013). In addition, the authors are sometimes vague as to what they mean by “antisociality.” They argue that antisociality predicts violence, based in part on papers that examined antisocial personality disorder in relation to violence. Violence in itself could be considered a manifestation of antisociality. Therefore, it is unclear how the researchers are defining this term.

• On page 3, the authors state, “The tendency to engage in antisocial behaviors such as violence, rule-breaking, and bullying varies significantly across the population, with a small proportion of individuals responsible for the majority of antisocial acts.” The citations used to justify this statement only apply to serious antisocial acts, rather than less serious antisocial acts that may be more evenly distributed in the population.

Methods and Results

• In the discussion, the authors argue that the sample roughly approximated major demographics of US adults ages 18-65. The authors should provide this demographic data so that the reader can judge for themselves the extent to which the sample was in fact representative.

• The theoretical basis for the classification of variables as predictors, outcome variables, or covariates is unclear. For example, why is PPE use frequency treated as a covariate rather than an outcome? Similarly, it seems that reported worry about COVID-19 might be an important covariate rather than an outcome variable.

• Participants came from 38 states. As disease spread varied widely by state in April 2020, including covariates that capture the extent of the state’s outbreak, as well as shutdown regulations is important. Income is another important covariate that is not included.

• The use of the silhouette image to assess social distancing is problematic. The image does not include any background images that might help the participant to gauge the relative size of the image. The authors interpret the image as being an average height male for their calculations, but this might not have been clear to the participant from the image. Also, why do the authors not take into account y-axis information when taking measurements? This would be relevant, as participants themselves had the ability to select distances based on both the x- and y-axes.

• Minor Point: Generally, correlation results are presented before regression results.

• The authors acknowledge this, but the lack of data on the reason for leaving home is a serious limitation. It could be the case that antisocial individuals had to leave the home for reasons that were not driven by their antisocial tendencies. Without data on the reasons for leaving home, it is difficult to derive causal inferences.

Discussion

• The authors state that 9.4% (3/32) of those who reported standing less than 6 feet from others also reported having flu-like symptoms. Given that this sample size is very small, it is not appropriate to draw conclusions or report results from these 3 participants.

• Answers to open-ended responses should not be reported in the discussion. These are results that should be reported in the results section and explained in the methods section.

6. PLOS authors have the option to publish the peer review history of their article (what does this mean?). If published, this will include your full peer review and any attached files.

Reviewer #1: No

Reviewer #2: No

Reviewer #3: No

---

## [Author Response · Author response to Decision Letter 0]

30 Sep 2020

Author response to the editor:

Thank you for your thorough and thoughtful comments. We believe our revised manuscript has substantially improved and now addresses all potential confounds, especially those related to economic factors by showing: 1) that removal of economic/financial items of the STAB measure did not affect results, 2) that including household income as a covariate does not affect results (while retaining covariates for demographics, education level and leaving the home for work), 3) that including participant-rated expected economic impact of COVID-19, does not affect results, and 4) that including the “left home for work” variable does not affect results relating antisociality to distance kept from others. We additionally explain why including “worry” as a covariate is inconsistent with our pre-registered hypotheses (but also show that including it as a covariate does not affect results).

We clarify a number of additional aspects of the manuscript. We now detail variations from the pre-registration and report analyses exactly as hypothesized in the supplementary material that show no changes to results. Throughout the paper, we also describe in more detail the context of the pandemic at the time of data collection and how this relates to our chosen dependent and independent variables. Most notably, we explain why PPE use frequency was not investigated as a dependent variable. It is important to recall that during the month preceding data collection, the public was urged not to purchase face masks because of shortages affecting healthcare workers (Amazon even temporarily prohibited public purchase of some PPE including N95 and surgical masks) and were even told that masks were not essential for preventing disease spread among non-healthcare workers. Although government guidelines were officially reversed to recommend face masks a few days prior to the study, availability was unequal and public messaging was inconsistent. Because of this, we anticipated the relationship between antisociality and PPE would be mostly uninterpretable, potentially because PPE use frequency would reflect a combination of access to resources/higher income, stockpiling against government officials’ recommendations, and concern about COVID-19. We therefore chose to focus on behaviors that violated clear and consistent COVID-19 guidelines, which requested people to stay home (stay-at-home orders affected 94% of Americans) and keep at least 6 feet of distance from others. We believe our results are meaningful in the context of the early phase of the pandemic and have also modified our title to clarify this.

We additionally clarify methods and variable measurements that reviewers noted as unclear or missing. Finally, we edited the manuscript to ensure that the magnitude of effect sizes are accurately contextualized and that results are not overstated. 

Additional specific comments for each reviewer are included in the Response to Reviewers PDF.

---

## [Decision Letter · Decision Letter 1]

3 Nov 2020

PONE-D-20-19373R1

Reduced social distancing early in the COVID-19 pandemic is associated with antisocial behaviors in an online United States sample

PLOS ONE

Dear Dr. O'Connell,

I am writing with a decision on your revised manuscript “Reduced social distancing early in the COVID-19 pandemic is associated with antisocial behaviors in an online United States sample.” I was fortunate to receive reviews from the same three scholars who reviewed the original manuscript, and I again reviewed the revised paper independently. As you can see, the reviewers felt you adequately addressed comments, and the paper is definitely strengthened and includes more nuanced interpretations. Reviewer 1 had a couple of remaining suggestions that would help integrate economic and ethnic/race explanations in the interpretation of results. I would also ask that you revise the manuscript to 1) reduce causal or “predictive” language, 2) make clearer that the MTurk sample doesn’t necessarily parallel US demographics, especially around income/education levels, and 3) acknowledge potential personality or other mechanisms that explain associations between antisociality and violations of COVID-related social distancing guidelines (relevant to previous concerns in the first round of reviews about addressing confounds and reducing overattribution of effects to antisociality). In particular, given the many personality traits and social factors that distinguish persons scoring high on a measure of antisocial behavior from other persons, what are possible reasons that these relationships are showing up as they are? For example, could extraversion (which can correlate with rule breaking or antisociality) be explaining the extent to which persons are leaving the house more often or standing closer to others – which would shift the implications of the findings in relation to risk factors and public health targets. I also noted that leaving home to go to work was also related to antisocial behavior, which again makes me wonder if there are potential social confounds or explanations that are not highlighted in explaining results. What could explain this connection? If you have data to address any of these, that would be important to consider. If the data are not available, the Discussion section could reflect alternative explanations instead of taking at face value that the tendency to not follow social norms is accounting for the results.

I believe that a revised manuscript that incorporates these minor comments would strengthen the contribution of the paper. Thank you for submitting your work for consideration in PLOS ONE.

We look forward to receiving your revised manuscript.

Kind regards,

Edelyn Verona

Academic Editor

PLOS ONE

Reviewers' comments:

Reviewer's Responses to Questions

**Comments to the Author**

1. If the authors have adequately addressed your comments raised in a previous round of review and you feel that this manuscript is now acceptable for publication, you may indicate that here to bypass the “Comments to the Author” section, enter your conflict of interest statement in the “Confidential to Editor” section, and submit your "Accept" recommendation.

Reviewer #1: All comments have been addressed

Reviewer #2: All comments have been addressed

Reviewer #3: All comments have been addressed

2. Is the manuscript technically sound, and do the data support the conclusions?

Reviewer #1: Yes

Reviewer #2: Yes

Reviewer #3: Yes

3. Has the statistical analysis been performed appropriately and rigorously? 

Reviewer #1: Yes

Reviewer #2: Yes

Reviewer #3: Yes

4. Have the authors made all data underlying the findings in their manuscript fully available?

Reviewer #1: Yes

Reviewer #2: Yes

Reviewer #3: Yes

5. Is the manuscript presented in an intelligible fashion and written in standard English?

Reviewer #1: Yes

Reviewer #2: Yes

Reviewer #3: Yes

6. Review Comments to the Author

Reviewer #1: The manuscript is a revised version of an article examining the relationship between antisocial traits as measured by the STAB and pandemic-related compliance with social distancing

and use of personal protective equipment (PPE, “mask, face shield and/or gloves”). I appreciated the author’s detailed attention in responding to the comments and re-organization of the manuscript which I agree strengthens their article.

With regard to my first concern, I think that the explanation in the manuscript is sufficient explanation and sufficiently addresses my concern.

With regard to my second related concern, I think that the author’s discussion in pages 398-402 should further clarify that their data seems to suggest at least two separate reasons for people leaving their home. The first one, possible antisociality and a second one, work. Therefore, rather than saying in p. 397-398 “Thus, leaving the home to work during the pandemic should not be considered an antisocial behavior” I would suggest changing to say something along the lines “Thus, it appears that antisociality may play a role in disregarding health guidelines in the pandemic for some people. However, for others, a prosocial motivation -employment- also independently influenced people’s decision to leave their houses.” I would disagree with the statement in line 402 saying “antisociality and reduced social distancing does not reflect these economic factors.” It seems more accurate to clearly state in the manuscript that both antisociality and need to go to work both independently contribute to leaving home especially since there was no interaction effect between these two variables.

I think that the authors’ hypotheses regarding the explanation for the paradox of higher education and less PPE is plausible and helps contextualize their findings in the time of the pandemic.

I think that the authors should include in the manuscript the Chi square analyses showing that Black and Latino respondents left for home more frequently and had higher scores on the economic items of the STAB and discuss these findings beyond saying that the findings should not be interpreted to suggest that going to work is antisocial behavior. The authors’ data and multiple other data clearly indicate that these ethnic backgrounds were unduly affected by the pandemic and forcing them to put themselves at risk to work more than other groups. The NPR story about the spread among the wealthy is irrelevant given that the wealthier around the world may contracting it more (arguably due to their antisocial disregard for self and others by holding soirees at fancy rose gardens despite a pandemic?) but are not forced to return to dangerous work conditions as members of these two ethnic groups have to do. In a period in which there is significant need to understand and highlight the role that racial and ethnic inequities in American society play in various health outcomes that point must be made given the authors’ data in this regard. Perhaps it should be synthesized with the discussion in lines 398-402/ the point I mentioned above.

I have no further concerns about this manuscript and I appreciate the opportunity to review this interesting study.

Reviewer #2: (No Response)

Reviewer #3: (No Response)

7. PLOS authors have the option to publish the peer review history of their article (what does this mean?). If published, this will include your full peer review and any attached files.

Reviewer #1: No

Reviewer #2: No

Reviewer #3: No

---

## [Author Response · Author response to Decision Letter 1]

18 Dec 2020

“1) reduce causal or “predictive” language”

We agree with the importance of using caution when it comes to causal language, and we have edited the manuscript to reduce causal language in the following places: line 348, line 370, line 410, line 470. Relatedly, we edited the abstract to reflect that sociodemographic variables were also associated with social distancing behavior.

“2) make clearer that the MTurk sample doesn’t necessarily parallel US demographics, especially around income/education levels”

We agree, and added the following sentence to the discussion section (lines 441-442), “MTurk samples are also not representative of U.S. sociodemographic variables and tend to include individuals with higher education and income [80]”

“3) acknowledge potential personality or other mechanisms that explain associations between antisociality and violations of COVID-related social distancing guidelines (relevant to previous concerns in the first round of reviews about addressing confounds and reducing overattribution of effects to antisociality). In particular, given the many personality traits and social factors that distinguish persons scoring high on a measure of antisocial behavior from other persons, what are possible reasons that these relationships are showing up as they are? For example, could extraversion (which can correlate with rule breaking or antisociality) be explaining the extent to which persons are leaving the house more often or standing closer to others – which would shift the implications of the findings in relation to risk factors and public health targets.”

We agree with the importance of acknowledging other potential factors. Lines 450-457 in the limitations now discuss this point. 

“I also noted that leaving home to go to work was also related to antisocial behavior, which again makes me wonder if there are potential social confounds or explanations that are not highlighted in explaining results. What could explain this connection?”…

This outcome was unexpected and has a variety of possible explanations, including the point above. For example, extraverted personality could tie together antisociality, social distancing and having a job that requires people to leave their home. Unfortunately, we do not have any additional data that could help explain this connection in any additional detail. We noted this point in our discussion of potential alternative explanations (lines 450-457). We also now highlight the association between socioeconomic variables and social distancing in the abstract (line 37-38).

Authors’ responses to Reviewer #1: 

“With regard to my second related concern, I think that the author’s discussion in pages 398-402 should further clarify that their data seems to suggest at least two separate reasons for people leaving their home…”

We appreciate the reviewer’s suggestion on how to clarify the results pertaining to subjects leaving for work. We have adapted the suggested wording to the manuscript (lines 400-407), but disagree with the framing of employment as a prosocial motivation—there are many reasons for going to work, only some of which would be described as prosocial.

“…I would disagree with the statement in line 402 saying ‘antisociality and reduced social distancing does not reflect these economic factors’…”

We have edited the manuscript (lines 409-412) to reframe the interpretation. The new interpretation indicates that the relationship between antisociality and reduced social distancing persists when adjusting for socioeconomic factors that independently contributed to reduced social distancing.

“…I think that the authors should include in the manuscript the Chi square analyses showing that Black and Latino respondents left for home more frequently and had higher scores on the economic items of the STAB and discuss these findings beyond saying that the findings should not be interpreted to suggest that going to work is antisocial behavior… In a period in which there is significant need to understand and highlight the role that racial and ethnic inequities in American society play in various health outcomes that point must be made given the authors’ data in this regard. Perhaps it should be synthesized with the discussion in lines 398-402/ the point I mentioned above.”

We now include the statistic that Black and Hispanic participants left for work more frequently (lines 312-315), included this result in the discussion (lines 404-407), and added three relevant citations to large, population-level studies that have been recently published that measure and report racial and ethnic disparities related to COVID-19 (line 407, references 74-76). We are confident that with regards to this topic, readers will be best served by data from large studies that are designed and sufficiently powered to address it.

We are concerned about including data about the STAB economic items for Black and Hispanic participants, however. For transparency, this would require including and interpreting data from the rest of the STAB as well, and devoting a significant proportion of the manuscript to interpreting these analyses, which were not pre-registered, which were not hypothesized a priori, and which the study is underpowered for. Moreover, in examining the subsample of 26 Black (n=15) and/or Hispanic (n=15) participants, we determined they are younger (Black and Hispanic: M=32.96, SD=1.93; Non-Black or Hispanic: M=37.17, SD=0.98; t(129)=1.93, p=.056) and more likely to have a 4-yr degree (Black and Hispanic=81%; Non-Black or Hispanic=50%; χ2=7.78, p=.005) than the rest of the sample. The very small size of these samples, the non-hypothesized nature of differences as a function of race, and the non-equivalence of Black and Latino participants compared to other participants on key variables indicates to us that such analyses (which, again, our study was not designed for, and which were not preregistered) would be scientifically misguided and the results of such analyses uninterpretable. Worse, the results could be easily misconstrued in a way that negatively reflects on minority individuals, which would be a significant cost, particularly given the ambiguity of these data.

---

## [Editor Report · Decision Letter 2]

21 Dec 2020

Reduced social distancing early in the COVID-19 pandemic is associated with antisocial behaviors in an online United States sample

PONE-D-20-19373R2

Dear Dr. O'Connell,

We’re pleased to inform you that your manuscript has been judged scientifically suitable for publication and will be formally accepted for publication once it meets all outstanding technical requirements.

Kind regards,

Edelyn Verona

Academic Editor

PLOS ONE
---

## [Editor Report · Acceptance letter]

28 Dec 2020

PONE-D-20-19373R2 

Reduced social distancing early in the COVID-19 pandemic is associated with antisocial behaviors in an online United States sample 

Dear Dr. O'Connell:

I'm pleased to inform you that your manuscript has been deemed suitable for publication in PLOS ONE. Congratulations! Your manuscript is now with our production department. 

Kind regards, 

on behalf of

Dr. Edelyn Verona 

Academic Editor

PLOS ONE